# Pre-Slavic and Slavic Interaction at Eastern Periphery of Slavic Expansion in Northeastern Europe (Y-Gene Pools of Volga-Oka Region)

**DOI:** 10.3390/genes16101149

**Published:** 2025-09-27

**Authors:** Dmitry Adamov, Alexsander Shlykov, Anna Potanina, Maria Voronina, Igor Gorin, Georgy Ponomarev, Danil Kabaev, Larisa Chernyaeva, Alexsander Gavrilov, Dmitry Rusakov, Elvira Pocheshkhova, Kristina Zhur, Egor Prokhortchouk, Natalia Goncharova, Elena Balanovska

**Affiliations:** 1Bochkov Research Centre for Medical Genetics, 115522 Moscow, Russia; shtrunov@yahoo.com (A.S.); potaninaaanna@gmail.com (A.P.); mybfisanihilist@yandex.ru (M.V.); gorin.io@phystech.edu (I.G.); st26i900@gmail.com (G.P.); darusakov93@ya.ru (D.R.); eapocheshkhova@mail.ru (E.P.); goncharovann@my.msu.ru (N.G.); balanovska@mail.ru (E.B.); 2Vladimir Regional Centre of Archeology, Vladimir State University, 600014 Vladimir, Russia; d.kabaev@mail.ru (D.K.); galchuk@list.ru (L.C.); 3Museum Historical and Cultural Complex of the Shilovsky Municipal District of the Ryazan Region, 391500 Shilovo, Russia; museum01@mail.ru; 4Department of Biology and Medical Technologies, Kuban State Medical University, Ministry of Health of the Russian Federation, 350063 Krasnodar, Russia; 5Federal Research Centre “Fundamentals of Biotechnology”, Russian Academy of Sciences, 119071 Moscow, Russia; zhur_kv@mail.ru (K.Z.); prokhortchouk@biengi.ac.ru (E.P.); 6Department of Anthropology, Faculty of Biology, Lomonosov Moscow State University, 119234 Moscow, Russia

**Keywords:** slavic expansion, pre-slavic, Y-chromosome, TMRCA, aDNA

## Abstract

**Background/Objectives**: The eastern periphery of the Slavic expansion (the Volga-Oka region) is the most promising region for reconstructing interactions between Slavic and pre-Slavic populations of the East European Plain. Unlike most pre-Slavic tribes, its autochthonous population practiced inhumation instead of cremation, leaving us with some ancient DNA for analysis. **Methods**: The region’s modern and ancient Y-chromosome gene pools are dominated by the haplogroup R1a: its frequency reaches 56% in Ryazan Russians (n = 302) and 44% in the Finnic peoples of Mordovia (n = 633). This encouraged us to analyze its Y-SNPs and Y-STRs. **Results**: Using 2 independent methods of phylogeny analysis, we identified 10 informative Y-STR clusters within R1a, dating back 1600–2900 YBP. The clusters included 48% of modern Ryazan Russians, 40% of Mordovia’s Finnic populations, and ancient DNA samples from the Ryazan-Oka culture (6–7th centuries), Suzdal (12–13th centuries) and Vladimir (13th century). Such a unique combination and pre-Slavic TMRCA indicate that the informative clusters represent pre-Slavic Y lineages. The presence of ancient samples from Vladimir and Suzdal in the clusters suggests that the autochthonous tribes contributed to shaping the urban population of the Vladimir-Suzdal Rus. Some of the informative clusters are associated with the ancient population of the Baltics (2000–4000 YBP). **Conclusions**: About half of Russian R1a carriers in the Volga-Oka region are descended from a pre-Slavic population, suggesting that the Slavs did not fully replace the autochthonous population but rather mostly culturally assimilated the Meshchyora documented in the Russian chronicles and other local tribes.

## 1. Introduction

Interactions between Slavic and pre-Slavic populations of the East European Plain have been studied by many disciplines. According to archaeological and toponymic evidence, Slavs had active contacts with the autochthonous tribes while expanding across the East European Plain. But due to methodological constraints, archaeology and linguistics cannot accurately assess the intensity of such contacts: the spread of a language or material culture does not always mark a change in a population’s makeup and can be associated with intertribal trade and linguistic infiltration. Biology provides almost no information about the physical appearance of the local pre-Slavic population or the rate of its change under the pressure of the Slavic expansion because cremation was the primary funeral custom in this region before Christianity. This is why genetic data is so important for reconstructing interactions between Slavic and pre-Slavic populations. The eastern periphery of the Slavic expansion is the most promising region for such reconstruction because, unlike most pre-Slavic populations, its inhabitants practiced in-ground burials, leaving us with some ancient DNA for analysis.

With no natural barriers, the East European Plain was a crossroads of multidirectional migrations; at the outset of the Slavic expansion in the 1st millennium CE, its population was very heterogenous due to the expansion of Baltic-speaking tribes from the West and Finno-Ugric tribes from the East (Appendix B). The arriving Slavs were heterogenous too: they carried traces of past interactions with Celtic and Germanic tribes [1]. In the 4–5th centuries CE, the Early Medieval Pessimum [2] pushed Slavs to migrate to the East European Plain; by the beginning of the 2nd millennium CE, the Slavic expansion had reached its easternmost frontier.

The autochthonous population from the eastern periphery of the Slavic expansion had been long absorbing migrations. The Gorodets culture that evolved from the Finno-Ugric Netted Ware culture and was influenced by the Proto-Balts existed in the Volga-Oka region until the 2nd century CE [3]. Notably, ancestors of Mordovia’s Erzya and Moksha emerged during the final stages of the Gorodets culture and within its geographical boundaries [4,5]. This makes the indigenous population of Mordovia a good model for reconstructing the pre-Slavic gene pool of the eastern periphery of the Slavic expansion. In the 2nd century CE, interactions between the descendants of the Gorodets culture, Balts and Sarmatians shaped the Ryazan-Oka culture (Appendix B) [6,7]. In the middle of the 1st millennium CE, it was heavily influenced by the Baltic tribes [1,4] displaced by early Slavic newcomers [1,4,8,9]. In the 7th century CE, the influx of steppe nomads split the Ryazan-Oka culture. Its descendants dispersed into enclaves that gave rise to the Meshchyora, other early medieval tribes and the gene pools of Mordovia’s Erzya and Moksha [10,11].

By the late 12th century, the Slavs had assimilated most of the autochthonous tribes in the Volga-Oka interfluve and founded the principalities of Ryazan and Vladimir-Suzdal [1,12]. The rate and character of the Slavic expansion varied by region. For example, the culture of the Finnic-speaking Muroma became extinct in the 12th century, but Finnic-speaking Meshchyora settlements in the historic Meshchyora region persisted until the 15–16th centuries because its woodlands and wetlands were taken up by the Slavs less intensively (Appendix B) [13,14].

By and large, the gene pool of the autochthonous population from the eastern periphery of the Slavic expansion was formed by Baltic and Finnic tribes, with some influence from steppe and Cis-Uralic tribes. Unlike most pre-Slavic populations, the pre-Slavic tribes of the Volga-Oka region (Muroma, Meshchyora, Proto-Mordvins) practiced inhumation in the 1st millennium CE, providing us with ancient DNA for analysis. This makes the Volga-Oka region the most promising model for studying interactions between Slavic and pre-Slavic populations of the East European Plain.

Previously, our team participated in an interdisciplinary study of linguistic data and autosomal and Y-chromosome gene pools [15] that demonstrated that all Slavic gene pools had a strong pre-Slavic substrate. The dramatic impact of pre-Slavic gene pools on the modern Slavic gene pool is indicated by 2 factors: (1) differences between East, West and South Slavs are determined by the differences between their pre-Slavic substrates; (2) the DNA markers are better correlated with geography than language.

Of all East Slavs, Russians have the most abundant and heterogenous gene pool. The diversity of Russian populations [16,17] is associated not only with the vastness of their habitat but also with the heterogeneity of the pre-Slavic substrate encountered by the Slavs on the East European Plain. It is the contribution of the autochthonous Baltic and Finnic populations that makes the gene pool of the Russian North so unique [18,19]. The pre-Slavic component represented by the Y haplogroups N3a3 and N3a4 constitutes a sizable proportion of the gene pool of Novgorod Russians [20]. The Y-chromosome gene pool of Yaroslavl Russians has a pre-Slavic substrate that might date back to the Merya people documented in historical chronicles [21].

Another study of autosomal gene pools confirms the important contribution of the pre-Slavic population to the genomic landscape of the East European Plain [22]. The ancestral ADMIXTURE components of Mordovia’s autosomal genomes (Erzya, Moksha and Shoksha) occur in most populations of the East European Plain regardless of their linguistic affiliation. The greatest contribution of the pre-Slavic ancestral ADMIXTURE component is observed in Ryazan Russians. This raises hope that Y-chromosome gene pools of indigenous Mordovian and Ryazan populations will provide evidence of interaction between Slavic and pre-Slavic populations.

Unfortunately, information about ancient DNA of the East European Plain is scarce. The stage of Russian ethnogenesis that preceded transition to the centralized Russian State in the 15th century is yet to be explored. The study of 32 samples from Suzdal Opolie dated to the 3rd–18th centuries pointed to the intensive assimilation of the autochthonous population by the Slavs [23]. When modeling the ancient genome of Alexander Nevsky’s son [24], the Shekshovo-9 genome from Suzdal Opolie [23] was interpreted as a result of interaction between Slavic and pre-Slavic components, and the ‘pure’ Slavic genome was modeled from a medieval Vladimir sample [25].

The aim of this study was to reconstruct the history of the gene pool at the eastern frontier of historical Russia by conducting a search for a pre-Slavic substrate in the modern population of the Volga-Oka interfluve. The reconstruction was based on the comparison of the gene pools of Ryazan Russians and Mordovia’s Finnic-speaking ethnicities using phylogenetic analysis, Y-SNP and Y-STR dating.

## 2. Materials and Methods

Samples (n = 935) were collected from the indigenous population of the Volga-Oka region during field expeditions in 2010–2023. Of them, 633 represented Mordovia (Ardatovsky, Chamzinsky, Ichalkovsky, Insarsky, Krasnoslobodsky, Lukoyanovsky, Ruzaevsky, Tengushevsky, Torbeyevsky districts) and 302 represented the Ryazan region (Kadomsky, Kasimovsky, Mikhaylovsky, Sapozhkovsky, Saraevsky, Shilovsky, Spassky districts). The samples were collected from unrelated men who gave voluntary written informed consent and whose ancestors from 2 previous generations represented the studied population. The samples were anonymized. The study was approved by the Ethics Committee of the Research Centre for Medical Genetics.

The gene pools of all Russian populations representing the Ryazan region were very similar (χ^2^) and hence analyzed as a single entity. Mordovian gene pools (Erzya, Moksha and Shoksha) differed significantly from each other (χ^2^) and were analyzed separately. Note that ‘Shoksha’ is an arbitrary name for the Erzyan population from Tengushevsky district with a distinct gene pool, language and culture [22,26].

DNA was extracted on magnetic beads using a Qiasymphony workstation (Qiagen, Hilden, Germany). Y-SNP genotyping included detailed R1a genotyping with Taqman Open Array custom plates on a QuantStudio 12 Flex real-time PCR platform (Thermo Fisher Scientific, Waltham, MA, USA). Samples of 401 R1a-Z280 carriers were analyzed for 37 Y-STR markers by fragment analysis (Appendix A) using a Nanofor-05 sequencing system (LLC Syntol, Moscow, Russia), commercial Yfiler Plus (Thermo Fisher Scientific, Waltham, MA, USA) and PowerPlex Y23 (Promega Corp., Fitchburg, WI, USA) kits and separate kits for DYS504, DYS525, DYS552, DYS505, DYS537, DYS445, Y-GATA-A10, and GGAAT1B07. Phylogenetic networks were constructed using a median-joining algorithm [27] implemented in Network v.10.2.0.0 and visualized in Network Publisher v.2.1.2.5. The weight and epsilon parameters were set to 10 and 0, respectively. TMRCA (time to the most recent common ancestor) estimates were calculated from Y-STR haplotypes of the modern samples using two independent methods: ASD and the rho-statistic with heuristic parsimony [19,28,29,30]. The average mutation rate for 37 Y-STRs was assumed to be 0.0039 per locus per generation [31,32]; the generation interval was assumed to be 31.5 years [33].

Distribution maps for the haplogroups R1a-Z92 and R1a-CTS1211 were built in GeneGeo v.2.8 [34] using the average weighted interpolation procedure with a search radius of 400 km and a weight function of 2. Composite portraits of Ryazan and Mordovian venous blood donors were created by superimposing and aligning the photos by reference points; the principle of equal contribution was observed for each photo [35,36].

Of 15 male aDNA samples representing the ancient Volga-Oka population, 5 belonged to R1a carriers. The sample aDNA-1 represented the Ryazan-Oka culture (Undrikh, the 6–8th centuries, the burial site of an upper-class warrior) [37,38]; aDNA-2 and aDNA-4 were from Vasilievsky burial site in Suzdal (the 12–13th centuries); aDNA-4 was from an earlier burial ground than aDNA-2. The samples aDNA-3 and aDNA-5 were from a ‘sanitary’ burial ground in the city of Vladimir, associated with the 1238 AD Mongol invasion.

## 3. Results

This section may be divided by subheadings. It should provide a concise and precise description of the experimental results, their interpretation, as well as the experimental conclusions that can be drawn.

### 3.1. Genetic Portraits of Modern Population at Eastern Periphery of Slavic Expansion (Y Haplogroups)

The genetic similarity between the gene pools of modern Slavic- and Finnic-speaking populations of the Volga-Oka region is manifest most visibly in the dominance of the major haplogroup R1a (48%, Table 1), making it the focus of this study. R1a, including its most common R1a-CTS1211 branch, prevails in all 3 Mordovia’s populations at 30–55% frequencies. R1a occurs at a higher frequency (56%) in Ryazan Russians, but is represented in this group by the R1a-Z92 branch more often than in Mordovians. R1a-CTS1211 and R1a-Z92 are the dominant R1a branches in all Volga-Oka populations, representing 86% of the R1a carriers.

Y haplogroup patterns differ significantly not only between the gene pools of Mordovia and the Ryazan region but also between 3 Mordovia’s populations (Table 1). The gene pools of the Erzya, Moksha and Shoksha are so remarkably different that they should be studied separately instead of being grouped together as the ‘Mordvin’. The Erzya are characterized by the prevalence of R1a (55%) and I1 (9%); the Moksha, by high E-M96 (17%) and J2-M172 (15%) frequencies; the Shokshan gene pool is dominated by N3a (45%). High R1a frequencies also occur outside the Volga-Oka region. R1a is a major, dominant haplogroup in East Slavs, constituting almost half (47%) of their gene pool, followed by I2 (14%) and N-M178 (12%). The gene pool of Russians follows the same pattern. The only region with 2 major haplogroups is the Russian North, where R1a occurs at 26% and N-M178 at 30% frequencies. In other Russian populations, R1a (53%) is strikingly more common than N-M178 (14%); other haplogroups occur at 1–7% frequencies.

Likewise, R1a prevails in the ancient population of the eastern periphery of the Slavic expansion: of 15 aDNA samples from the Undrikh burial site, Suzdal and Vladimir (the 6–13th centuries), 5 (33%) represent R1a carriers. The dominant position of R1a in the gene pool of the Vladimir-Suzdal Rus is confirmed by [23]: of 8 aDNA samples, 3 (38%) represented R1a. So, it would be reasonable to first look for the genetic traces of pre-Slavs within R1a, which has prevailed in the region since antiquity.

The distribution map of the most widespread R1a branches (R1a-CTS1211 and R1a-Z92) shows the position of Mordovia’s and Ryazan’s gene pools among the indigenous gene pools of the entire East European Plain (Figure 1; the boundaries of the regions are shown in white). R1a-CTS1211 occurs at high frequences in the Volga-Oka interfluve (Figure 1A), reaching its peak (47%) in the Ryazan Russians of Kasimovsky district (No. 32 on the map). High R1a-CTS1211 frequencies are typical for South Russians, declining eastward to the Cis-Urals and the Russian North. R1a-Z92 (Figure 1B) is less common and has a peak frequency in the west of the East European Plain among Tver, Yaroslavl and Orel Russians. Its frequency predictably declines eastward, reaching zero values in the Cis- Urals. The same “west-to-east” pattern is observed for R1a-Z92 in the Volga-Oka region, where it has the highest frequency in the westernmost Russian populations and the lowest frequency in the easternmost populations of Mordovia.

### 3.2. Search for Pre-Slavic Genetic Substrate Using R1a Y-STR Data

It would be reasonable to start searching for a pre-Slavic substrate among those R1a branches that are the most typical for the Volga-Oka region, i.e., R1a-CTS1211 and R1a-Z92. If we find such Y-STR haplotypes that emerged before the arrival of Slavs (TMRCA > 1000 years) and occur simultaneously in Ryazan Russians and Mordovia’s Finnic-speaking peoples, then such Y-STR haplotypes will indicate the presence of a pre-Slavic substrate in the modern Russian gene pool.

Phylogenetic networks of Y-STR haplotypes and their phylogenetic trees for R1a-CTS1211 and R1a-Z92 were constructed on the basis of 401 Y-STR haplotypes representing the modern indigenous population of the Volga-Oka region: 154 Russian samples from 7 districts of the Ryazan region and 247 samples from Mordovia (148 Erzyans, 72 Mokshans, 27 Shokhans). The identified Y-STR clusters corresponded well with Y-SNP variants of the haplogroup R1a (Appendix A).

Five major mutations Y3910, Y33 (xY1390), Y35 (xY33), CTS1211 (xY35), and Z92 divided the Y-STR haplotypes into 5 more or less equally sized groups, for which 5 ‘search’ phylogenetic networks (Appendix A) and 24 ‘search’ phylogenetic trees (Appendix A) were constructed. They were searched for such informative clusters of Y-STR haplotypes that simultaneously met two requirements: (1) they included individuals from modern Slavic (Ryazan Russians) and Finnic-speaking (Mordovians) populations; (2) they dated back before the Slavic colonization (before the 10th century). If any of the requirements was not met, the cluster was considered non-informative and was excluded from the analysis.

Ten informative clusters of pre-Slavic Y-STR haplotypes were identified (Appendix A). Importantly, these clusters included not only Russians but also individuals from 3 Mordovia’s populations: 46% of Mokshans, 44% of Shokshans, 32% of Erzyans. This suggests that the pre-Slavic substrate is ‘universal’ for the Volga-Oka region. Moreover, 10 informative clusters containing pre-Slavic haplotypes included half (74 of 154!) of all Russian R1a carriers. In other words, the proportion of descendants from the pre-Slavic population among today’s Russian R1a carriers is 48% at the eastern periphery of the Slavic expansion. Since R1a frequency in Ryazan Russians is 56% (Table 1), over one-fourth (27%) of indigenous Russians in the Ryazan region are descended from a pre-Slavic population. Note that this paper analyzes the haplogroup R1a only; the search for a pre-Slavic substrate among other haplogroups deserves a separate publication.

The phylogenetic networks and trees for 3 most widespread R1a subbranches with pre-Slavic Y-STR haplotypes included ancient DNA samples (Figure 2, Figure 3 and Figure 4).

The sample aDNA-1, which dates back before the Slavic expansion (the 6–7th centuries) and was recovered from the Undrikh burial site in the Ryazan region, appears in the R1a-Y35(xY33) network (Figure 2A), confirming the pre-Slavic origin of lineages in this network. The R1a-Y35(xY33) network demonstrates that 23 Russians from 7 Ryazan districts and 13 Erzyans, Mokshans and Shokshans from Mordovia have common pre-Slavic ancestors (1720–2880 YBP). The Erzyan and Russian branches in the R1a-YP335 cluster (Figure 2B) diverged 1000 years ago.

The R1a-Z92 network (Figure 3A) contains aDNA samples from Suzdal (aDNA-2) and Vladimir (aDNA-3), indicating the presence of pre-Slavic Y-chromosome lineages in the urban population of the Vladimir-Suzdal Rus (12–13th centuries). The R1a-Z92 network shows that 26 Russians from 7 Ryazan districts and 21 Erzyans, Mokshans and Shokshans from Mordovia have common pre-Slavic ancestors (2030–2410 YBP).

In the R1a-Y33 network (Figure 4), 25 Russians from 5 Ryazan districts and 59 Erzyans, Mokshans and Shokshans from Mordovia have common pre-Slavic ancestors (1600–2410 YBP).

Thus, all informative clusters dated to 1600–2880 YBP comprise Russians (n = 74) from all the districts of the Ryazan region and individuals from all 3 Finnic-speaking populations of Mordovia (n = 93). Indeed, TMRCA estimates were calculated from the modern Y-STR haplotypes. But the fact that ancient samples (aDNA-1, aDNA-2, aDNA-3) are found within three of the phylogenetically defined clusters validates TMRCA estimates from modern genomes. The oldest aDNA-1 sample from the Undrikh burial (Ryazan-Oka culture, the 6–7th centuries CE) is direct evidence of the pre-Slavic origin of cluster YP335 (Figure 2). The samples aDNA-2 from Suzdal (pre-Mongol Rus, 12–13th centuries CE) and aDNA-3 from Vladimir (the first third of the 13th century) show that the autochthonous pre-Slavic population was incorporated into the gene pool of the urban population of the Vladimir-Suzdal Rus (Figure 3); this finding is supported by archaeological [39] and autosomal genome [23] data. Indeed, a larger ancient DNA sample size is needed to estimate the amount of such incorporation. The samples DNA-4 (Suzdal, the 12–13th centuries) and aDNA-5 (Vladimir, the first third of the 13th century) lie outside the informative clusters.

## 4. Discussion

Our findings are the first proof of a significant pre-Slavic input into the modern Russian gene pool of the Volga-Oka region. About half (48%) of our R1a Russian carriers represent those R1a lineages that were spread in the region before the Slavic expansion. Considering the high frequency of R1a, at least one-fourth (27%) of its Russian carriers in the region are descended from a pre-Slavic population. This means that the Slavs did not fully replace the autochthonous population but rather culturally assimilated the Meshchyora, Muroma and other pre-Slavic tribes. This is why the pre-Slavic contribution to the modern Russian gene pool is so substantial.

The significant pre-Slavic input into the modern gene pool that formed at the eastern periphery of the Slavic expansion is confirmed by the high frequency of the same R1a lineages in the indigenous population of Mordovia: about 40% of this population share common pre-Slavic ancestors with Ryazan Russians. This confirms that Russians and Mordovians have common pre-Slavic ancestors.

Another important factor is the spread of pre-Slavic R1a branches in the region. First, there are 10 such branches, and this tells against the hypothesis that the high proportion of Russians descended from pre-Slavic populations might be associated with strong genetic drift in the carriers of only one pre-Slavic R1a branch. Second, pre-Slavic R1a branches are uniformly distributed across all districts of the Ryazan region. The same uniformity is observed for all Mordovia’s populations (Erzya, Moksha and Shoksha), confirming that these R1a branches were widespread in the entire pre-Slavic population of the Volga-Oka region.

Evidence from humanities suggests that at the eastern periphery of their expansion Slavs encountered the early medieval Finnic-speaking Meshchyora, Muroma, Mordvin and other tribes, whose names are unknown. Their autochthonism is confirmed by historical continuity: from Netted Ware to Gorodets to Ryazan-Oka to early medieval tribes. Of all Finnic-speaking peoples, only Mordvinic tribes retained their identity. They have given rise to the modern Erzya, Moksha and Shoksha and are a link to the ancient pre-Slavic population.

The strength of our conclusions rests on the careful selection of blood donors: the samples were collected from rural populations that follow the tradition of monoethnic marriages and represented unrelated individuals whose grandfathers belonged, by birth, to the studied population and identified likewise. Evidence from ethnography, archaeology and history points to insignificant miscegenation between Mordovia’s indigenous populations and Russians. Therefore, it is very unlikely that almost half (44%) of the Mordovians are descended from Russians. This conclusion is indirectly supported by pre-Slavic TMRCA, the continuous passage of archaeological cultures and the gradual emergence of Slavic elements in the Meshchyora and Muroma cultures. Importantly, the results of Y-chromosome data analysis reflect the presence of the autochthonous male component in the general population besides the Slavic component from men who engaged in marriages with local women. Previous studies of autosomal genomes and mtDNA [15,22] also suggest a substantial pre-Slavic contribution to the gene pool of East Slavs.

The most important confirmation of the strong pre-Slavic input into the Russian gene pool comes from the dating of 10 informative clusters. Three independent dating methods produced similar results (Table 2); the lowest estimate was 1560 ± 230 YBP and the highest 2880 ± 540 YBP (YFull estimates were slightly higher due to the inclusion of lineages outside of the Volga-Oka region). All dating techniques show that all 10 informative R1a branches existed in the region long before the arrival of Slavs. If Slavs had carried the same R1a lineages before their expansion as the autochthonous population, then, considering that there are 10 informative clusters dated to 1600–2900 YBP, we would have to assume either close contacts between Slavs and the autochthonous population long before (>1000 years) Slavic colonization or a shared ancestral population. However, these assumptions are not supported by archaeological evidence.

The origins of pre-Slavic R1a branches can be reconstructed from archaeological data and aDNA. According to radiocarbon dating, R1a-Z93, typical for the representatives of the Fatyanovo culture (Bronze Age pastoralists, presumably early Indo-European speakers), dominated the Upper and Middle Volga regions 4500–4200 YBP [40,41]. R1a-Z280 emerged 500 years later (3700 YBP) [42] and gave rise to R1a-CTS1211 and R1a-Z92 common for the Volga-Oka region (Table 1 and Table 2). Between 4000 and 2000 YBP, these R1a branches were widespread in the Baltic region: of 13 informative samples, 3 represented R1a-Z92 and 10 represented R1a-CTS1211 [43,44] (Appendix A). The oldest R1a-CTS1211 (~3900 years) was found in Lithuania [43]. The samples from Lithuania also carried R1a-YP617 (≈2900 YBP) [43], the ancestral branch for clusters Y11268 and YP683 that are informative for the Volga-Oka region (Appendix A). The sample from Estonia (2500 YBP) [44] lies on the S4481subbranch, and so does the Volga-Oka cluster R1a-Z92 (Appendix A). This gene geographic pattern is consistent with a potential Baltic trace: some of the identified informative R1a lineages may have shared a common ancestor with Proto-Baltic or other Indo-European tribes living to the west of the Volga-Oka interfluve [4]. Further ancient DNA evidence is needed to distinguish between a direct migration from the Baltic zone and a shared common ancestry from earlier times.

In any case, all R1a lineages dated to 4000–1000 YBP bear no relation to the Slavic expansion but might be associated with Proto-Baltic tribes. All age estimates for the informative Volga-Oka R1a branches fall within this interval (the oldest branch dates back 2880 ± 540 YBP, the youngest 1560 ± 230 YBP) and indicate a significant contribution of the autochthonous pre-Slavic population to the modern Russian gene pool of the Volga-Oka region.

Summing up, this comprehensive study of Y-chromosome gene pools from the eastern periphery of the Slavic expansion has revealed genetic traces of the autochthonous population in the modern Russian population of the Volga-Oka region and in the ancient urban population of Vladimir and Suzdal. Inclusion of the indigenous population in the dataset and the careful choice of Mordovian populations as a model of a pre-Slavic population ensures the reliability of the obtained results. Our findings suggest that the Slavic population of this region was formed through mostly cultural assimilation (Slavicisation) of the local Finnic-speaking tribes (Meshchyora and their early medieval kin tribes) but not through their complete replacement.

## 5. Conclusions

1. R1a dominates the Y-chromosome gene pools (n = 935) of the modern population of the Volga-Oka region: it is found in 56% of Ryazan Russians and 44% of Mordovia’s indigenous populations. The Y-chromosome gene pools of the Erzya and Moksha are so different from each other that they should be analyzed separately instead of being grouped together as the Mordvin.

2. Using two independent methods for phylogeny reconstruction, we were able to detect 10 haplogroup R1a sublineages of possible pre-Slavic origin descendants in the modern population of the Volga-Oka region dated from 2880 ± 540 YBP to 1560 ± 230 YBP.

3. The frequency of the pre-Slavic substrate lineages is high: about half of the Russian R1a carriers (i.e., one-fourth of the Russian population) are descended from the pre-Slavic population of the region. This suggests that the Russian population of the Volga-Oka interfluve was largely formed by the cultural assimilation rather than by the complete replacement of the Meshchyora and other extinct autochthonous tribes by the Slavs.

4. The analysis of ancient DNA samples from the Ryazan-Oka culture (the 6–7th centuries CE), Suzdal (late 12–early 13th centuries) and Vladimir (the first third of the 13th century) shows that the autochthonous pre-Slavic population was incorporated into the gene pool of the urban population of the Vladimir-Suzdal Rus. A larger ancient DNA sample size is needed to estimate the amount of such incorporation.

5. Many pre-Slavic R1a branches date back over 2500 years and are phylogenetically related to the ancient population of the Baltic region (2000–4000 YBP).

## Figures and Tables

**Figure 1 genes-16-01149-f001:**
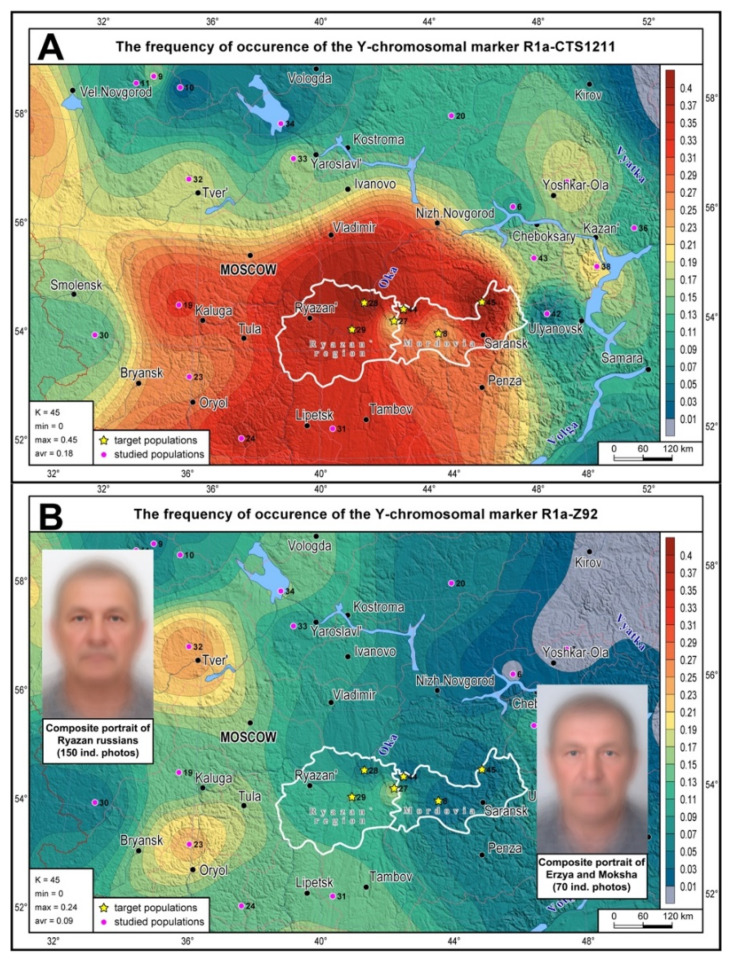
The distribution of haplogroups R1a-CTS1211 (**A**) and R1a-Z92 (**B**) across indigenous populations of the East European Pain. The studied populations are marked with pink dots; yellow stars represent target populations of the Ryazan region and Mordovia (their administrative boundaries are highlighted in white). The color bar legend for haplogroup frequencies is provided in the right panel. The figure also shows composite portraits of (1) Ryazan Russians and (2) Erzyans and Mokshans (**B**).

**Figure 2 genes-16-01149-f002:**
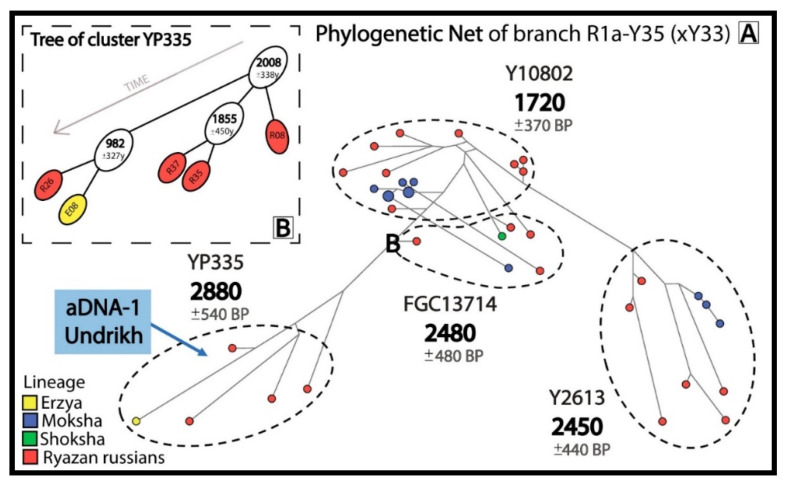
Informative clusters of probable descendants of the pre-Slavic population in the phylogenetic network for R1a-Y35(xY33) (**A**) and a phylogenetic tree for cluster R1a-YP335 (**B**).

**Figure 3 genes-16-01149-f003:**
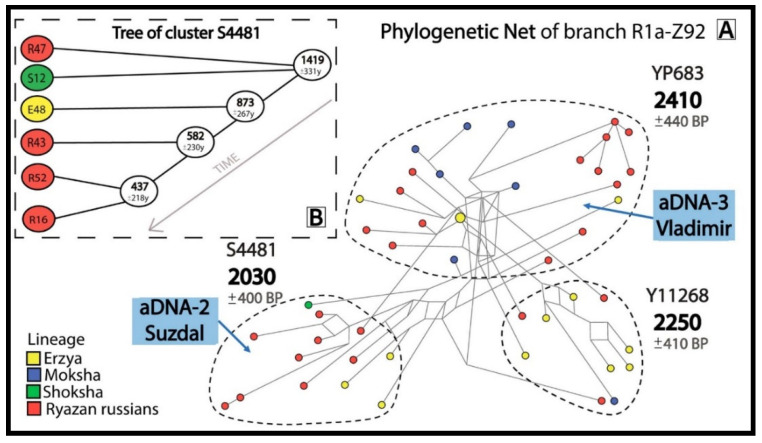
Informative clusters of probable descendants of the pre-Slavic population in the phylogenetic network for R1a-Z92 (**A**) and a phylogenetic tree for cluster R1a-S4481 (**B**).

**Figure 4 genes-16-01149-f004:**
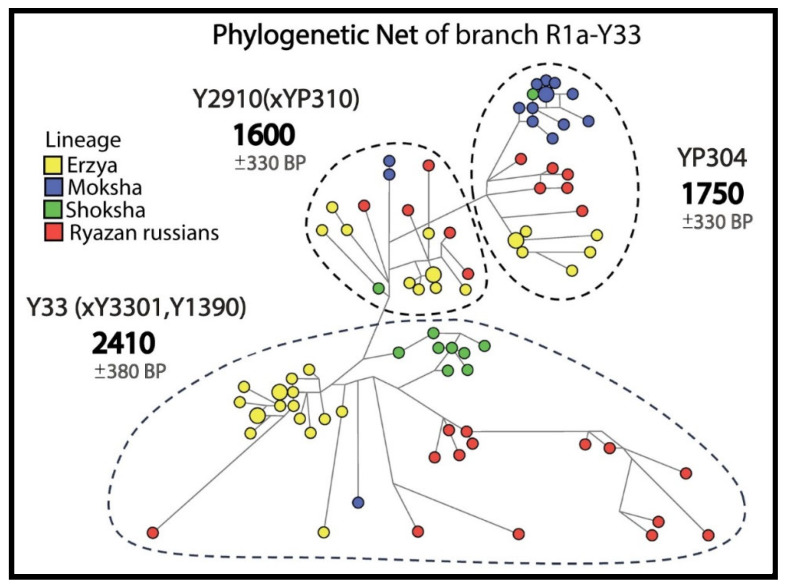
Informative clusters of probable descendants of the pre-Slavic population in the phylogenetic network for R1a-Y33.

**Table 1 genes-16-01149-t001:** Frequencies of Y-chromosome haplogroups (%) in the indigenous populations of the Volga-Oka region (Russians from the east of the Ryazan region and Mordovia’s Erzya, Shoksha and Moksha).

Y-Haplogroups	Pooled Populations			Ethnic Groups of Mordovia		
	Volga-Oka Region	Ryazan Russians	Mordvins	Erzya	Moksha	Shoksha
**Number of individuals**	913	280	633	317	251	65
**R1a (total)**	47.9	56.4	44.1	55.5	30.3	41.5
R1a-CTS1211	35	35	35.1	423	24.7	40
R1a-Z92	6.2	11.8	3.8	4.1	4	1.5
R1a-PF6202 (eq.M458)	3.4	7.9	1.4	2.5	0.4	0
R1a-Z93	3.2	1.8	3.8	6.6	1.2	0
N3 (total)	15	11.4	16.6	13.2	13.5	44.6
N3a1-B211	7.2	0.4	10.3	8.8	8.4	24.6
N3a3-CTS10760 (eq.VL29)	4.2	6.4	3.2	0.6	4.8	9.2
N3a4-Z1936	3.6	4.6	3.2	3.8	0.4	10.8
**Haplogroups (xR1a. N3)**	37.1	32.1	39.3	31.2	56.2	13.8
E-M78	6.6	2.5	8.4	2.8	16.7	3.1
I2-P37.2	3.3	9.3	0.6	0.9	0.4	0
J2-M172(xM67)	6.1	0.7	8.5	4.7	15.1	1.5
I1-M253	4.3	3.6	4.6	8.5	0.8	0
R1b-M269 (xZ2105)	4.3	2.1	5.2	3.5	7.2	6.2
G2-P303	4.4	1.8	5.5	3.8	9.2	0
Other	8.2	12.1	6.5	6.9	6.8	3.1

Note. Data meeting the 5% allele frequency criterion is highlighted in bright red. Haplogroups are arranged in the descending order of their frequencies in the gene pool of the Volga-Oka region. “Shoksha” is an arbitrary name for the Erzyan population from Tengushevsky district in the north-west of Mordovia.

**Table 2 genes-16-01149-t002:** Age of 10 informative R1a branches (years BP) estimated by 3 different methods.

R1a Branch	Number of Individuals	TMRCA, ASD, Our Own Data	TMRCA, Rho-Statistic, Our Own Data	YFull, SNP, Data from Yfull
YP335	5	2880 ± 540	2010 ± 340	2100 ± 200
Y2613	9	2450 ± 440	1580 ± 280	2600 ± 300
Y10802	16	1720 ± 370	1620 ± 320	1800 ± 190
FGC13714(xYP236, YP954, Y9118)	6	2480 ± 480	2150 ± 380	3600 ± 230
S4481	13	2030 ± 400	1780 ± 270	3600 ± 380
Y683	24	2410 ± 440	1560 ± 230	2000 ± 330
Y11268	10	2250 ± 410	1590 ± 310	2000 ± 310
Y2910(xYP310)	18	1600 ± 330	1670 ± 350	1950 ± 160
YP304	26	1750 ± 330	1760 ± 350	2300 ± 160
Y33(xY3301, Y1390)	40	2410 ± 380	2410 ± 330	4400 ± 330

**Legend:** TMRCA is time to the most recent common ancestor.

## Data Availability

Data and materials are available in Appendix A.

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
