# Peer review of "Pre-Slavic and Slavic Interaction at Eastern Periphery of Slavic Expansion in Northeastern Europe (Y-Gene Pools of Volga-Oka Region)"

_genes, 2025, doi:10.3390/genes16101149_

Round 1

Reviewer 1 Report

Comments and Suggestions for Authors

The manuscript “Pre-Slavic and Slavic Interaction at eastern periphery of Slavic expansion in Northeastern Europe (Y-gene pools of Volga-Oka region)” quantifies the proportion of the R1a haplogroup among the Y chromosomes of populations with putative pre-slavic ancestry and characterizes its phylogenetic history. Specifically, it dates the major branching events of the haplogroup and concludes that they are mostly dated before the Slavic expansion. These conclusions are sound.

My only major comment is that while I understand that the study focuses on a specific haplogroup in the sample of pre-slavic populations, I think it would have been more complete by addressing the possibility that some of these R1a lineages might have also been present in Slavic populations, thereby potentially overestimating the pre-slavic contribution. This could be addressed by including other populations with Slavic ancestry, including ancient ones. Maybe this is already clear to the authors and that is why they did not think it is was necessary to add, but it is not obvious for the reader and it sounds like an obvious missing piece to the puzzle. I would suggest the authors to add something in this respect. The fact that an haplogroup has an origin pre-dating the Slavic expansion is not sufficient in itself – as it could pre-date the expansion but also be shared with the expanding population.

Minor comments:

In several places, the text could receive more care.

It is repeated several times (at least three in abstract/introduction) that because of inhumation the target populations are ideal. It is OK, it is a central concept. But I would avoid at least the repetition in line 89.

At the beginning of the Results it says:

This section may be divided by subheadings. It should provide a concise and precise

171

description of the experimental results, their interpretation, as well as the experimental

172

conclusions that can be drawn.

This is clearly some leftover from AI generated text. I think it is OK to write or polish the paper with ChatGPT. But at least remove such phrases, please.

Here and there, there are strange wordings. Like “8makes” in line 101. I would suggest a thorough read-through of the manuscript.

Some fixes are still necessary, particularly in regard to the inclusion/comparison with slavic ancestry and polishing of the text.

Reviewer 2 Report

Comments and Suggestions for Authors

This article presents a significant and well-structured study on the genetic history of the Volga-Oka region, offering valuable insights into the nature of the Slavic expansion in Northeastern Europe. The research is ambitious, combining modern and ancient DNA analysis with archaeological and historical context. However, while its conclusions are compelling, several methodological and interpretive aspects should be critically examined. Therefore, I enclose below some suggestion in order to improve the article, for which I congratulate the authors.

Page 4. Lines 159-162. Why are relevant the composite portraits? Please, explain what relation they have with the study.

Page 8, Line 263: "Note that this paper analyzes the haplogroup R1a only..."

The authors should explicitly acknowledge in the Discussion (Page 10 or 12) that the Y-chromosome provides only a patrilineal perspective. They should state that their conclusions about assimilation, while strongly supported by this line of evidence, would be significantly strengthened by future studies incorporating autosomal DNA and mitochondrial DNA to capture a more complete demographic history, including female-mediated gene flow and overall ancestry proportions.

Page 8, Lines 245-265 (Method for identifying informative clusters).

The identification of "informative clusters" is based on two criteria: 1) presence in both Slavic and Finnic speakers, and 2) a TMRCA predating the Slavic expansion. However, the TMRCA dating itself is calculated from the modern samples. While the inclusion of aDNA (like the Undrikh sample) helps break this circularity for specific clusters, the method for initially defining all clusters risks presupposing the result it seeks to find.

Thus, for improving the text, the authors should add a sentence to the Methods section (Page 4, Paragraph 2) or the Results section (Page 8) clarifying the process. In my view, it should be make explicit that for clusters without supporting ancient DNA, the pre-Slavic designation is based on the TMRCA estimate calculated from the modern haplotypes. However, the subsequent finding of ancient samples (aDNA-1, aDNA-2, aDNA-3) within three of these phylogenetically-defined clusters validates this methodological approach for those specific lineages.

Page 10, Line 299: "...the urban gene pool of the Vladimir-Suzdal Rus actively incorporated the autochthonous pre-Slavic population..."

Given the very small sample size (3 aDNA samples in clusters), the conclusion should be framed more cautiously. The sentence could be rephrased in the Discussion, indicating that the contribution to the urban gene pool of the Vladimir-Suzdal Rus, while suggestive, needs to be confirmed with a larger ancient DNA sample size in order to determine the amount of incorporation. Small size is a problem in most of the ancient DNA studies. Please, drop a line to moderate conclusions or ideas when there are only a few samples.

Page 11, Lines 342 and further (mention of confidence intervals and YFull estimates).

The authors should include the confidence intervals (e.g., ± value) for the TMRCA estimates of the 10 informative clusters directly in the main text (Table 2 or in the narrative on Page 11) rather than just stating they were considered. This would allow readers to better evaluate the robustness of the dating, especially for the younger clusters (~1560 YBP) that are closer to the Slavic expansion timeline.

Page 11, Lines 367 and further: "This suggests the Baltic trace..."

The language in the Discussion should be moderated from suggestive to more cautious. It must be clear that the phylogeographic pattern is consistent with a potential Baltic trace, suggesting that some of these informative R1a lineages may share a common ancestor with populations located in the Baltic region during the Bronze and Iron Ages (as the authors suggest). But it must be explicited that further ancient DNA evidence is required to distinguish between a direct migration from the Baltic zone and a shared common ancestry from an earlier period.

Page 12, Lines 391-394. Confidence intervals should be taken into account. Writting should be moderated.

Page 12, Lines 401-405. The 4th conclussion should be rewritten: the authors only have 3 samples (one for each site)! Something indicating that the authors have this limitation in mind must be included. In my view, the whole conclusión section should moderate the strength of the claims. This does not make the article worse but turns the text more realistic and, therefore, improves the scientifical contribution.

Other issues:

For those unfamiliar with russian geography, a general B/W schematic map should be included in the text (not as supplementary info) comprising all the regions and locations relevant (also the rivers).  Figure 1 is not quite clear. Being more pedagogic would help reading and understanding of the text. In the introduction a lot of regions, tribes and locations are mentioned and the reader does not have anything “at hand”. If you are not familiar with the region it is highly difficult to follow the explanation from the text. This map should be referred also in the apendix section.

Minor issues:

Please, be sure that characters from S6-S11 can be seen when printed. Font size is really small.

Author Response

Response to Reviewer 2 Comments

1. Summary

Thank you very much for taking the time to review this manuscript. Please find the detailed responses below and the corresponding revisions/corrections highlighted in red.

2. Questions for General Evaluation

Reviewer’s Evaluation

Response and Revisions

Does the introduction provide sufficient background and include all relevant references?

Yes

Agree

Are all the cited references relevant to the research?

Yes

Agree

Is the research design appropriate?

Can be improved

Agree

Are the methods adequately described?

Yes

Agree

Are the results clearly presented?

Can be improved

Agree

Are the conclusions supported by the results?

Yes

Agree

3. Point-by-point response to Comments and Suggestions for Authors

Comments 1: Page 4. Lines 159-162. Why are relevant the composite portraits? Please, explain what relation they have with the study.

Response 1: Thank you for your keen attention to the visuals. The composite portraits are provided to give the readers an idea of the anthropological type of the donors. We would like to emphasize that the portraits are based on the images of only those individuals whose Y chromosomes are studied in this paper. These portraits refute the popular saying "Scratch a Russian and you will find a Tatar". They demonstrate that Finnic-speaking and Slavic populations of the region are close in terms of their anthropological types.

Comments 2: Page 8, Line 263: "Note that this paper analyzes the haplogroup R1a only..."

The authors should explicitly acknowledge in the Discussion (Page 10 or 12) that the Y-chromosome provides only a patrilineal perspective. They should state that their conclusions about assimilation, while strongly supported by this line of evidence, would be significantly strengthened by future studies incorporating autosomal DNA and mitochondrial DNA to capture a more complete demographic history, including female-mediated gene flow and overall ancestry proportions.

Response 2: Thank you for pointing this out.

Previously, we studied the autosomal genomes and mtDNA of Slavic populations and came to the conclusion about a substantial pre-Slavic substrate. In-text citations of these publications are provided in lines 91–96 and 106–113 of this manuscript. Consequently, we write about our intention to study other Y haplogroups in lines 264–265.

Per your advice, we have added the following clarification to the Discussion section:

“Importantly, the results of Y-chromosome data analysis reflect the presence of the autochthonous male component in the general population besides the Slavic component from men who engaged in marriages with local women. Previous studies of autosomal genomes and mtDNA [15,22] also suggest a substantial pre-Slavic contribution to the gene pool of East Slavs.”

Comments 3: Page 8, Lines 245-265 (Method for identifying informative clusters).

The identification of "informative clusters" is based on two criteria: 1) presence in both Slavic and Finnic speakers, and 2) a TMRCA predating the Slavic expansion. However, the TMRCA dating itself is calculated from the modern samples. While the inclusion of aDNA (like the Undrikh sample) helps break this circularity for specific clusters, the method for initially defining all clusters risks presupposing the result it seeks to find.

Thus, for improving the text, the authors should add a sentence to the Methods section (Page 4, Paragraph 2) or the Results section (Page 8) clarifying the process. In my view, it should be make explicit that for clusters without supporting ancient DNA, the pre-Slavic designation is based on the TMRCA estimate calculated from the modern haplotypes. However, the subsequent finding of ancient samples (aDNA-1, aDNA-2, aDNA-3) within three of these phylogenetically-defined clusters validates this methodological approach for those specific lineages.

Response 3: Thank you for your suggestion about refining the description of the methods used in the study.

The following changes have been made to the Materials and Methods section (page 4, lines 152–154):

“TMRCA (time to the most recent common ancestor) estimates were calculated from Y-STR haplotypes of the modern samples using two independent methods: ASD and the rho-statistic with heuristic parsimony [19, 28–30].”

The following sentences have been added to the Results section (page 9, lines 194–195):

“Indeed, TMRCA estimates were calculated from the modern Y-STR haplotypes. But the fact that ancient samples (aDNA-1, aDNA-2, aDNA-3) are found within three of the phylogenetically defined clusters validates TMRCA estimates from modern genomes.”

Comments 4: Page 10, Line 299: "...the urban gene pool of the Vladimir-Suzdal Rus actively incorporated the autochthonous pre-Slavic population..."

Given the very small sample size (3 aDNA samples in clusters), the conclusion should be framed more cautiously. The sentence could be rephrased in the Discussion, indicating that the contribution to the urban gene pool of the Vladimir-Suzdal Rus, while suggestive, needs to be confirmed with a larger ancient DNA sample size in order to determine the amount of incorporation. Small size is a problem in most of the ancient DNA studies. Please, drop a line to moderate conclusions or ideas when there are only a few samples.

Response 4: We appreciate your remark. The sentence in the Results section on page 10 (lines 297–301) has been amended to read as follows:

“The samples aDNA-2 from Suzdal (pre-Mongol Rus, 12th–13th centuries CE) and aDNA-3 from Vladimir (the first third of the 13th century) show that the autochthonous pre-Slavic population was incorporated into the gene pool of the urban population of the Vladimir-Suzdal Rus (Fig.3); this finding is supported by archaeological [30] and autosomal genome [31] data. Indeed, a larger ancient DNA sample size is needed to estimate the amount of such incorporation.”

Comments 5: Page 11, Lines 342 and further (mention of confidence intervals and YFull estimates).

The authors should include the confidence intervals (e.g., ± value) for the TMRCA estimates of the 10 informative clusters directly in the main text (Table 2 or in the narrative on Page 11) rather than just stating they were considered. This would allow readers to better evaluate the robustness of the dating, especially for the younger clusters (~1560 YBP) that are closer to the Slavic expansion timeline.

Response 5: Thank you for pointing this out.

The sentence on page 11 (lines 344–346) has been amended to read as follows:

“Three independent dating methods produced similar results (Table 2); the lowest estimate was 1560 ± 230 YBP and the highest 2880 ± 540 YBP (YFull estimates were slightly higher due to the inclusion of lineages outside of the Volga-Oka region).”

Comments 6: Page 11, Lines 367 and further: "This suggests the Baltic trace..."

The language in the Discussion should be moderated from suggestive to more cautious. It must be clear that the phylogeographic pattern is consistent with a potential Baltic trace, suggesting that some of these informative R1a lineages may share a common ancestor with populations located in the Baltic region during the Bronze and Iron Ages (as the authors suggest). But it must be explicited that further ancient DNA evidence is required to distinguish between a direct migration from the Baltic zone and a shared common ancestry from an earlier period.

Response 6: Following your advice, we have changed the tone of the sentences on pages 11 and 12 (lines 367-372) from suggestive to more cautious:

“This gene geographic pattern is consistent with a potential Baltic trace: some of the identified informative R1a lineages may have shared a common ancestor with Proto-Baltic or other Indo-European tribes living to the west of the Volga-Oka interfluve [4]. Further ancient DNA evidence is needed to distinguish between a direct migration from the Baltic zone and a shared common ancestry from earlier times.

In any case, all R1a lineages dated to 4000–1000 YВР bear no relation to the Slavic expansion but might be associated with Proto-Baltic tribes.”

Comments 7: Page 12, Lines 391-394. Confidence intervals should be taken into account. Writting should be moderated.

Response 7: Following your advice, we have changed paragraph 2 in the Conclusions section (page 12, lines 391–394) to read as follows:

“2. Using two independent methods for phylogeny reconstruction, we were able to detect 10 haplogroup R1a sublineages of possible pre-Slavic origin descendants in the modern population of the Volga-Oka region dated from 2880 ± 540 YВР to 1560 ± 230 YВР.”

Comments 8: Page 12, Lines 401-405. The 4th conclussion should be rewritten: the authors only have 3 samples (one for each site)! Something indicating that the authors have this limitation in mind must be included. In my view, the whole conclusión section should moderate the strength of the claims. This does not make the article worse but turns the text more realistic and, therefore, improves the scientifical contribution.

Response 8: Thank you for your comment.

Paragraph 4 of the Conclusions section (page 12, lines 401–405) has been reworded:

“4. The analysis of ancient DNA samples from the Ryazan-Oka culture (the 6th–7th centuries CE), Suzdal (late 12th–early 13th centuries) and Vladimir (the first third of the 13th century) shows that the autochthonous pre-Slavic population was incorporated into the gene pool of the urban population of the Vladimir-Suzdal Rus. A larger ancient DNA sample size is needed to estimate the amount of such incorporation.”

Comments 9: Other issues:

For those unfamiliar with russian geography, a general B/W schematic map should be included in the text (not as supplementary info) comprising all the regions and locations relevant (also the rivers).  Figure 1 is not quite clear. Being more pedagogic would help reading and understanding of the text. In the introduction a lot of regions, tribes and locations are mentioned and the reader does not have anything “at hand”. If you are not familiar with the region it is highly difficult to follow the explanation from the text. This map should be referred also in the apendix section.

Response 9: Thank you for the valuable suggestion. The map is now included in the Appendix A.

Comments 10: Minor issues: Please, be sure that characters from S6-S11 can be seen when printed. Font size is really small.

Response 10: Thank you for pointing this out.

It is difficult to change the font size, but the quality of Supplementary Figures S6-S15 has been considerably improved. Smaller details can be seen by zooming in on the text.

4. Response to Comments on the Quality of English Language

Point 1: (x) The English is fine and does not require any improvement.

Response 1: Agree

5. Additional clarifications

No response